# Normative Data for Ten Neuropsychological Tests for the Guatemalan Pediatric Population Updated to Account for Vulnerability

**DOI:** 10.3390/brainsci11070842

**Published:** 2021-06-25

**Authors:** Joaquín A. Ibáñez-Alfonso, Rosalba Company-Córdoba, Claudia García de la Cadena, Ian C. Simpson, Diego Rivera, Antonio Sianes

**Affiliations:** 1Human Neuroscience Lab, Department of Psychology, Universidad Loyola Andalucía, 41704 Sevilla, Spain; jaibanez@uloyola.es (J.A.I.-A.); rcompany@uloyola.es (R.C.-C.); 2ETEA Foundation, Development Institute of Universidad Loyola Andalucía, 14004 Córdoba, Spain; 3Department of Neuropsychology, University of the Valley of Guatemala, Guatemala City 01015, Guatemala; claudigd@uvg.edu.gt; 4Department of Health Sciences, Public University of Navarre, 31006 Pamplona, Spain; diegoriveraps@gmail.com; 5Research Institute on Policies for Social Transformation, Universidad Loyola Andalucía, 14004 Córdoba, Spain; asianes@uloyola.es

**Keywords:** cognitive assessment, neuropsychology, vulnerability, development, cross-cultural standardization, 2030 Agenda, normative data, disadvantaged

## Abstract

The Guatemalan pediatric population is affected by a high incidence of poverty and violence. The previous literature showed that these experiences may ultimately impact cognitive performance. The aim of this article is to update the standardized scores for ten neuropsychological tests commonly used in Guatemala considering vulnerability. A total of 347 healthy children and adolescents from 6 to 17 years of age (*M* = 10.83, *SD* = 3) were assessed, controlling for intelligence, mental health and neuropsychological history. The standard scores were created using multiple linear regression and standard deviations from residual values. The predictors included were the following: age, age squared (age^2^), mean parental education (MPE), mean parental education squared (MPE^2^), gender, and vulnerability, as well as their interaction. The vulnerability status was significant in the scores for language, attention and executive functions. To the best of our knowledge, this is the first study that includes the condition of vulnerability in the calculation of neuropsychological standard scores. The utility of this update is to help in the early detection of special needs in this disadvantaged population, promoting more accurate interventions in order to alleviate the negative effects that living in vulnerable conditions has on children and adolescents.

## 1. Introduction

According to the American Psychological Association [1], child neuropsychology is the discipline that studies children’s behavior connected to brain structures and functions from infancy to adolescence. Differing from adult neuropsychology, child neuropsychology recently emerged [2], and is focused on by many clinical professionals. Thus, child neuropsychology seeks to understand and to measure whether neurobehavioral processes are being successfully fulfilled during different developmental stages. The mental processes that may help an individual receive, select and manage information from the self and from the context are known as cognitive functions. The proper assessment of these functions helps to track the adequacy of childhood development and to ensure attention to special needs. Commonly, these functions are the following: perception, praxia, memory, attention, language, and executive functions. The degree to which these neuropsychological processes are studied in pediatric populations varies from country to country, and the interest in them is increasing in Spanish-speaking populations [3]. Specifically, according to the data offered by Arango-Lasprilla et al. [4], 33.9% of the clinical neuropsychology professionals in Latin America and 15.8% of the clinical professionals in Spain work with children and adolescents. One of the limitations that these professionals experience is the scarcity of proper standards for the Latino population. For example, Oliveras-Rentas et al. [5] conducted a study on the state of the neuropsychology profession in Spanish-speaking countries. One of their findings was that professionals often had to use standards normalized for other populations to assess their patients, even though they had doubts as to the applicability of this practice, due to the scarcity of appropriate normative data.

A key concept in clinical neuropsychological work is the use of standards established using normative data. According to Mitrushina et al. [6], among other criteria, standards “must be readily available to the professional community and adequately normed”. These authors highlighted the relevance of sociocultural determinants in the adequacy of these measures. Standards are conventionally established values for evaluating something or someone with regards to a specific topic. The standards are used to interpret neuropsychological tasks since they allow clinicians to understand the position in terms of the performance of a specific child with respect to other children, taking into account his or her personal determinants (e.g., age, gender). As Anastasi stated [7], assessing individuals from a cultural group using data normalized for another cultural group can lead to false pathological scores. Nevertheless, many clinical professionals use neuropsychological tests without the proper standards to compare their pediatric patients [8]. Despite the use of normative data being widespread for years, it was not until 1991 that variables other than age (e.g., gender or education) began to be included in the standardization of neuropsychological testing standards [9]. Scores on cognitive tests are presumably influenced by personal factors, such as age, gender, education, parent’s educational level, and many other factors that may influence children’s and adolescents’ cognitive performance. Accordingly, it is important to consider these individual characteristics in order to interpret the neuropsychological test scores. The need to develop appropriate and validated standards for minorities or specific cultural groups is a cause for concern for specialist authors [10,11,12]. Some authors have recently stated the necessity of having appropriate standards to assess cognitive functions since cultural experiences (e.g., traditions or beliefs) may interfere with, or influence, neuropsychological performance [13]. Within Latino/Spanish-speaking populations, authors such as Echemendía and Julian [12] have stated the necessity of developing standards appropriately adapted to specific cultural groups, considering the group traits. These authors also stated that although the translation of neuropsychological tests is a key step in the field, much more effort is needed for the proper development of cross-cultural neuropsychology in this sense.

According to Arango-Lasprilla et al. [4], some of the more commonly used neuropsychological tests in the Spanish-speaking pediatric population are the following: the Boston Naming test, the California Verbal Learning test, the Digit and Symbols test, the Peabody Picture Vocabulary test, the Rey Complex Figure test, the Stroop Color and Word Interference test, the Trail Making test, the Verbal Fluency test and the Wisconsin Card Sorting test. Thus, in 2017, an international team of researchers led by Arango-Lasprilla et al. [14] developed a methodology to standardize 10 of the most commonly used neuropsychological tools in clinically healthy Spanish-speaking children and adolescents from ten Latin American countries (Chile, Colombia, Cuba, Ecuador, Guatemala, Honduras, Mexico, Paraguay, Peru, and Puerto Rico) along with Spain. All of these tests are well described in the literature and are commonly used to assess the cognitive functions of children and adolescents. The normative data produced by this process are published [14].

The present article focuses on Guatemala, which was one of the countries included in the development of the normative data. This country is characterized, among other particularities, by important cultural diversity, largely due to the number of different indigenous populations that coexist. With regards to socioeconomic factors, although Guatemala possesses the largest economy of the Central American region [15], its socioeconomic difficulties are evident due to the scarcity of resources and existing violence in some regions of the country. Moreover, a high proportion of the Guatemalan population is young, with children under 14 years of age representing 34.55% of the total population [16]. Moreover, the children and adolescents of Guatemala have to face diverse difficulties on a daily basis, including community violence. Thus, infancy in this country is characterized by hazards mainly related to the familiar poverty status and exposure to violence.

Consequently, these children and adolescents develop in disadvantaged and complex contexts that generate instability. One way in which these life conditions affect younger populations is by increasing their stress levels. Elevated levels of stress have been shown to have a detrimental effect on human brain functioning. For example, experiencing stress decreases brain alpha-wave frequencies, and increases blood pressure and pulse rate [17], and consequently has a negative impact in the human body, influencing personal wellbeing as well as cognitive performance. If living in chronic stress conditions significantly affects adults’ wellbeing, for children, those effects are even worse. Specifically, the way in which stress affects the development of children of disadvantaged conditions was studied [18,19,20]. The mechanism by which stress affects the human brain is through the dysregulation of the hypothalamic–pituitary–adrenal axis, increasing cortisol levels. These changes negatively impact prefrontal areas, which are especially involved in executive processes.

Recent studies have examined the magnitude of differences in performance on neuropsychological tasks between vulnerable and nonvulnerable children and adolescent populations [21]. Based on their results, the authors of this article consider it necessary to adapt the current standards to include the vulnerability status of the assessed population. Diverse studies already offer information about how vulnerability affects children’s development through the structural and functional impacts on an infant’s brain [22,23,24], and as a consequence, performance in the main cognitive processes are affected (for review, see Ibáñez-Alfonso et al. [21]).

The existing differences between groups in terms of vulnerability are remarkable, and the inclusion of vulnerability in the regression model of each cognitive task is necessary since this condition may significantly interfere with children’s and adolescents’ cognitive performance. According to the Royal Academy of the Spanish Language, the term *vulnerable* refers to the quality of someone being hurt or receiving a physical or mental injury [25]. In this sense, children who live in vulnerable conditions due to the level of violence surrounding them and/or the scarcity of resources due to employment conditions, educational level or family income generated by parents are considered a vulnerable population. Despite this, as commonly socioeconomic indexes are used to capture these social groups’ realities, it is necessary to understand the complexity of these contexts. As Duncan and Magnuson recommended [26], the socioeconomic status is such a multifaced concept that is difficult to summarize it in a single index. For this reason, using qualitative data in order to determine children and adolescents’ vulnerability (for instance, reported neighborhood conditions, violence experienced, etc.) should be also considered.

Consequently, it is easy to imagine that, as vulnerable children have special conditions that they face on a daily basis (e.g., undernutrition, crowded households, lack of stimulation, difficulties in family attachment, and possible trauma), they should not be assessed using the same standards as children who do not live with these conditions. Thus, measuring children’s cognitive capacities with standards created using nonvulnerable populations leaves thousands of children and adolescents in a disadvantaged situation, as they are compared with a group that does not belong to their normal context. Accordingly, the adjustment of normative data in neuropsychological tests is needed, especially for minorities and groups with life conditions which make them particularly vulnerable. 

In this regard, local and international organizations have to intervene in order to guarantee the most vulnerable segments of populations’ well-being. In order to do so, the United Nations considered the inclusion of the Sustainable Developmental Goals (SDGs) in the 2030 Agenda. This action tends to promote equality in terms of opportunities for all citizens, with special emphasis on those goals that aim to alleviate poverty and exclusion in disadvantaged groups of populations [27], as is the case for children and adolescents. Since leaving no one behind is the main premise of the SDGs, actions implemented in order to achieve these goals have to be implemented from diverse fields. The inclusion of disadvantaged groups in the upgrade of the neuropsychological standards is a specific action that promotes not forgetting that these children exist and that, due to their context, they require special attention from different areas of study.

Thus, the aim of this study is to offer an upgrade of the previously existing normative data in 10 neuropsychological tests for the pediatric Guatemalan population, adding more participants to the previous standards and including the condition of suffering from vulnerability as part of the update.

## 2. Materials and Methods

### 2.1. Participants

In this study, two groups were considered: nonvulnerable and vulnerable. The sample was composed of 347 healthy children from Guatemala with ages ranging from 6 to 17 years old (*M* = 10.8, *SD* = 3, 47.8% females). The mean years of parental education was 8.8 (*SD* = 4.7). The initial sample from which standard measures were created included 203 nonvulnerable participants (age *M* = 11.2, *SD* = 2.5, 46.8% females), and a cohort of 144 vulnerable participants (age *M* = 11.0, *SD* = 3.7, 49.3% females) was added in order to update the standards. The age ranges’ description is included in Appendix A. The nonvulnerable participants were students from middle-low urban contexts from the city of Guatemala, and the vulnerable participants lived in economically deprived and especially violent rural and urban areas. The sample was selected by convenience. Specifically, the schools selected for inclusion in the study were based on the expert advice of the Fe y Alegria Foundation, an international organization dedicated to improving the social conditions of the impoverished and excluded sectors of society. The vulnerability condition was determined by the children and adolescents’ context (violence and poverty). Chronic exposure to risks is a key concept for understanding vulnerability. These risks can be related to, for instance, contextual difficulties regarding the economy, health hazards and personal insecurity factors [28]. In the present study, the mean monthly family income for the vulnerable sample was GTQ 1782 (*SD* = GTQ 1739.7, mean equivalent of approximately USD 232.50). To put this into context, the mean rent per capita in Guatemala was approximately GTQ 2980 in 2019 [29].

Those included in the sample met the following inclusion criteria: (a) being from 6 to 17 years old, (b,) having the Spanish language as a first or proficient second language, (c) having an IQ ≥ 80 according to the Test of Nonverbal Intelligence TONI-2 [30], (d) having a score < 19 on the Children’s Depression Inventory (CDI) [31], and (e) having a score ≤ 65 in the typical scores of the subscale of general anxiety of the Revised Child Anxiety and Depression Scale (RCADS) [32]. Participants were excluded under the following conditions: (a) having a history of disease in the central nervous system that can interact with neuropsychological disturbances (e.g., head trauma or epilepsy), (b) having medical records of any psychiatric disorder (e.g., major depression or bipolar mood disorder), (c) having sensory deficits that could make the assessment process difficult (e.g., vision problems or hearing loss), (d) having a diagnosed neurodevelopmental disorder, (e) having a score > 5 on the Alcohol Use Disorders Identification Test (AUDIT-C) [33] for participants 12 years of age and older, (f) having consumed psychoactive substances in the previous 6 months for participants 12 years old and older, and (g) having a history of prenatal problems (e.g., hypoxia or spina bifida).

### 2.2. Procedure

Parental informed consent for inclusion in the study was obtained for all participants prior to commencement. The study was conducted in accordance with the Declaration of Helsinki, and the protocol coded be443f9fda0f49abcfe25cbda85ce6e5e3444776 was approved by the Research Ethics Committee of the Andalusian government on 6 April 2019. Additionally, it was supported by the Research Ethics Committee of the Universidad del Valle de Guatemala. The project on which this study is based is indeed based on a first project that aimed to create standards for neuropsychological tasks in the Latin-American and Spanish infant population. So, first, a research group conducted a project to create standards for 10 neuropsychological tests for Latin-American and Spanish populations [14]. In the samples used to create the norms, there was a higher number of younger children (around 6 years old) than adolescents. Subsequently, we conducted a project focused on the Guatemalan population, which is characterized by high violence and poverty rates [21]. The term “update” in this case stands for the fact of upgrading the previously existing data for the Guatemalan population by adding more adolescents to the sample and by considering the vulnerability of the sample. The assessment procedures in both the standardization study [14] and the updating study were previously described elsewhere. The assessments in the standardization study took place from July to October 2016, and the assessments of the updating study took place from March to June 2019.

### 2.3. Instruments

#### 2.3.1. Screening Tests

Clinical and sociodemographic interview for participants’ parents: Questions regarding information about different areas of the children and adolescents’ health status were answered by the parents. Thus, a questionnaire included the following information: data about pregnancy, complications in childbirth, sensory difficulties and mental or physical medical complications. To determine the socioeconomic characteristics of the assessed sample, the parents completed a questionnaire that contained information such as the number of inhabitants at home, income per month, education level and job status.

Nonverbal intelligence: This skill was assessed using the TONI-2 [30]. The concurrent validity of this test was assessed with the Wechsler Intelligence Scale for Children—3rd edition, WISC-III [34], which showed good concurrent validity with other WISC-III subtests [35,36]. This test assesses IQ with minimum language or culture influence and evaluates problem solving skills as a general construct to measure intelligence through 55 items with increasing difficulty. TONI-2 is recommended to be used with people from 5 to 85 years old. The participants must complete a sequence of abstract shapes and select a final shape from a set of alternatives. The raw scores can be transformed into IQ scores and transformed into percentiles. Standardized scores under 80 were considered very low, so participants who scored below this threshold were excluded. Completing the test takes approximately 15–20 min.

Depression: Scores related to depressive symptomatology were assessed with the Spanish version [37] of the Children Depression Inventory, CDI [31]. The CDI has shown good psychometric properties with a test–retest reliability measured by Cronbach’s α of 0.84 [38] and 0.74 in children from Guatemala [39]. This self-reported inventory uses 27 items to assess the depressive symptomatology experienced by children and adolescents from 7 to 17 years old. Each item can be answered on a three-point scale of three, where 0 = lack of symptoms, 1 = moderate symptoms and 2 = severe symptoms, meaning that higher scores indicate higher depression symptomatology. The final score is the sum of the responses to each item, with 54 being the maximum possible score and scores ≥ 19 considered to be cause for concern. Participants who obtained scores above this threshold were excluded from the study, due to the possible existence of depression. Completing this test takes approximately 15 min.

Anxiety: Anxiety symptomatology was assessed using the General Anxiety Subscale of the Revised Child Anxiety and Depression Scale, RCADS [32]. Good psychometric properties, such as internal consistency across countries and different languages [40], and adequate internal consistency in the Guatemalan pediatric population [41] were shown by this scale. This study used a subscale of 6 items answered on a scale ranging from 0 to 3, and questions on her/his feelings were assessed only for 3rd grade and above children. Participants who scored ≥ 65 were excluded, due to the possibility of having anxiety symptomatology. This subscale takes 5 min to complete.

Alcohol consumption: Alcohol intake was measured using the modified Alcohol Use Disorders Identification Test, AUDIT-C [33]. The complete version of the AUDIT showed good psychometric properties in terms of reliability and validity, compared with the Michigan Alcoholism Screening test MAST [42], which showed a good correlation (*r* = 0.88) [43]. In this questionnaire, adolescents (≥12 years old) were asked about their alcohol consumption in the last 6 months. This version has 3 items with scores that can vary from 0 to 4. Obtaining a score above 5 on this scale was an exclusion criterion. This questionnaire takes approximately 4 min to complete.

Psychoactive substance consumption: This was assessed using a checklist for psychoactive substances. This short test included questions about psychoactive substance intake in the last 6 months, and it was administered to adolescents 12 years of age or older. More commonly consumed psychoactive substances were included in the alternatives of consumption (e.g., tobacco or cannabis) and other psychoactive substances (e.g., heroin or cocaine). A participant was directly excluded from the study if they reported the consumption of one substance in the previous 6 months. This checklist takes 5 min to complete.

#### 2.3.2. Neuropsychological Tests

The updated standards correspond to 10 commonly used neuropsychological tests, recently standardized in Guatemalan pediatric populations. In neuropsychology, it is difficult to find a test that exclusively assesses a certain cognitive domain [44]. In order to obtain a comprehensive view of how cognitive processes work in these children and adolescents, we chose tests that assess praxia, memory, language and executive function processes. These tests were used in a previous study that laid the foundation for the current work [21].

(a)Rey Osterrieth Complex Figure Test (ROCF) [45]:

The ROCF test assesses multiple skills, such as memory, motor, perceptive and attention [45]. This test can be used to assess the level of figure activity development or to evaluate the state of the perceptive component in neurologic disorders. This task also assesses executive processes, such as planning and problem solving [46]. For this test, the participant had to copy a complex, meaningless figure and draw it later. Then, two scores were obtained from this test, i.e., copy and memory, with a maximum score of 36.

(b)Learning and Verbal Memory Test (TAMV-I) [47]:

The TAMV-I test offers different verbal memory scores: immediate, delayed and recognition. The test consists of three parts: (1) verbal repetition, which serves as the learning phase; (2) delayed memory; and (3) visual recognition. In the learning phase, the evaluator read a word list that contained 12 words from three categories: clothing, furniture, and body parts. The list was repeated 4 times, and the participants were required to repeat the list each time. Prior to commencing the verbal repetition tasks, the participants were informed that there would be a subsequent memory test of these items later in the session. Thirty minutes later, a delayed memory task was applied. Finally, there was a recognition task in which the participants saw a list of words and had to indicate which of these were in the initial test. The maximum score in the first subtest is 48, and 12 is the maximum score in the other two subtests.

(c)Symbol Digit Modalities Test (SDMT) [48]:

This is a classic test that allows the rapid detection of brain dysfunctions through a task of substituting symbols for digits. Although it is a simple task, it involves the unification of complex neurophysiological processes that underlie visual, attentional, executive, motor and language functions. This test is used in the education field to assess reading skills and to predict reading disturbances in children [49]. This task consists of replacing figures with numbers by using a previously learned key. The final score is the total number of correct substitutions made in 90 s. The maximum score is 110.

(d)Concentration Endurance Test (d2) [50]:

The d2 test mainly assesses attention selective processes by using a cancelation task [50]. By using this test, the assessor obtains scores of processing speed, instruction commitment and execution in visual stimuli discrimination. The d2 test assesses the relationship between speed and accuracy of attention. The d2 test can be applied to children, adolescents, and adults. The test has 14 lines with 47 items each. The participant has to mark the three items that the examiner previously showed. The participants have 20 s to complete each line. Although the instrument has different scores (e.g., total responses, total correct responses, omission and commission errors, effectiveness of the performance, and concentration index), in this study, only the concentration index was used, due to its relevance to the research objectives.

(e)Peabody Picture Vocabulary Test (PPVT-III) [51]:

The PPVT-III test is commonly used in order to assess receptive vocabulary and as a verbal skill screening task [51]. The PPVT-III is used to assess receptive vocabulary and can be used as a screening method to obtain a quick view of verbal attitudes in individuals aged 2 to 90 years. The tool contains 192 items in total, each consisting of a sheet with 4 black and white illustrations. The task is to select the illustration that best fits the meaning of a word that is verbally presented by the examiner. Test administration begins with age-appropriate items.

(f)Shortened version of the Token test [52]:

The reduced version of the Token test enables the assessment of verbal order comprehension without redundancy, with these orders increasing in complexity. The tool includes 20 tokens in five colors, two sizes, and two shapes and the answer document. This reduced tool has 36 items of increasing difficulty, which are divided into six parts. Those items are orders that the participants must complete (e.g., touch the small red circle and the big green square). The maximum score in this task is 36.

(g)Phonological and Semantic Verbal Fluency Test [53]:

This test was administered and scored following the guidelines of Olabarrieta-Landa et al. [54]. The test includes two different verbal fluency modalities: phonological and semantic. The number of words emitted in each category in 60 s is measured and is considered a controlled verbal production test. In the phonological subtest, children had to say as many words as they knew, starting with /f/, /a/, and /s/. In the semantical subtest, children had to produce words belonging to two categories: animals and fruits.

(h)Stroop Color and Word Interference Test [55]:

This test assesses processes related to cognitive flexibility and resistance to interference [55]. This test is composed of three pages with a total of 100 items in each one. The first part has “red”, “green”, and “blue” written on it in black ink. The second part has 100 elements “xxxx” inked with different colors (red, green, or blue). The last part has the same number of elements, but this time, they are color words inked with mixed colors. The score is the number of items that the participant can read in 45 s for each trial. The three types of scores can be used to calculate reading speed, color denomination speed, and inhibition control ability, respectively.

(i)Trail Making Test A-B (TMT A-B) [56]:

This tool assesses processes related to search and visual exploration, processing speed, cognitive flexibility and general executive functions [56]. The TMT is a very popular diagnostic tool used with diverse types of cerebral dysfunction. The participant must connect dots in a logical order, with the TMT-A being composed of numbers (increased order), and the TMT-B composed of numbers and letters (alternating numerical increasing order and alphabetical order). Part A provides information about motor, visual–spatial search and sustained attention skills. Part B measures alternating attention and is a great measure of cognitive flexibility [57].

(j)Modified Wisconsin Card Sorting Test (M-WCST) [58]:

The M-WCST is used to assess problem-solving strategies, abstract thinking and cognitive flexibility, which are core processes of executive functions [58,59]. This version has a total of 48 response cards and four key cards. The examiner takes the first category chosen by the participant as correct. The examiner only informs of whether the election is correct or not, and when the rule has changed at the sixth correct classification. When the participant keeps classifying cards according to the previous category after the rule has changed and the examiner has informed that it is an incorrect response, it is considered a perseverative error. The test ends when six categories are correctly classified or until the participant has used all cards. The number of correct categories completed (max. 6) and the number of perseverative errors is usually taken to measure problem-solving ability and cognitive flexibility, respectively.

### 2.4. Data Analysis

The analyses were completed using SPSS version 26 [60]. Missing values were coded, using different numerical items in order to indicate the reason for the lack of information (e.g., not applicable, “not reading competent”, and “not numerically competent”). After codification, the authors conducted an exploratory analysis in order to detect outliers in the raw scores, which were treated using the winsorization method [61], considering values ±3 *SD* as the cutoff points. An independent samples *t*-test revealed a non-significant difference of 0.3 years in the age of the vulnerable and nonvulnerable cohorts (95% CI—4, 1.0; *t* [235.5] < 1, *p* = 0.400). The mean parental education was significantly different, favoring the parents from the nonvulnerable group (vulnerable: *M* = 6.5, *SD* = 4.48; nonvulnerable: *M* = 10.5, *SD* = 4.01, *t* [3 45] = −8.62, *p* = 0.001). A chi-squared test revealed no significant difference in terms of the gender distribution between the two cohorts (χ^2^ [1] = 0.212, *p* = 0.645).

### 2.5. Adjusted Standardization

Unlike the process used to create the first standards [14], in the new standardization process, the vulnerability variable was included (dummy coded: vulnerable = 0 and nonvulnerable = 1), and the mean of the parents’ education years was treated as a continuous variable. Gender was also coded, where female = 0 and male = 1. Multiple regression models of each task were obtained based on the following: age, age^2^, mean of parental education (MPE), MPE^2^, gender, vulnerability and interactions among these variables. Age and the mean years of parental schooling were centered in order to avoid multicollinearity [62]. The Bonferroni correction was applied to the *p* value due to the resulting number of comparisons among variables, with this value rounded up (*p* = 0.005). The final models were obtained using a hierarchical elimination process. The model tested for all scores was as follows (predicted value for neuropsychological scores based on regression models):(1)$${\hat{Y}}_{i}=B_{0}+B_{1}\cdot {\left(Age-{\bar{X}}_{Age}\right)}_{i}+B_{2}\cdot {\left(Age-{\bar{X}}_{Age}\;\right)}_{i}^{2}+B_{3}\cdot {\left(MPE-{\bar{X}}_{MPE}\right)}_{i}\;+B_{4}\cdot {\left(MPE-{\bar{X}}_{MPE}\right)}_{i}^{2}+B_{5}\cdot Gender_{i}+B_{6}\cdot Vulnerability_{i}+B_{K}\cdot Interactions_{i}+e_{i}$$

The interactions included all two-way combinations between each term of the remaining factors, such as *B_k_*$\cdot {\left[\left(\mathrm{Age}-{\bar{X}}_{Age}\;\right)\cdot \left(\mathrm{MPE}-{\bar{X}}_{MPE}\right)\right]}_{i}$. In the final models, no variable that would be included in a higher-order predictor was eliminated.

Through the following four-step procedure [63], a final predictive model was obtained by using the previously offered equation. (Step 1) After conducting the hierarchical elimination process, the selection of the significant parameters (B) generated a model for each neuropsychological task. Using the final regression model for each task, the predictive value was obtained $\left({\hat{Y}}_{i}\right)$ (formula 1). (Step 2) Then, the residual value of the model was obtained $\left(e_{i}\right)$ by subtracting the raw score $\left(Y_{i}\right)$ from the predictive value of each case in each task. Therefore, the formula was $e_{i}=Y_{i}-{\hat{Y}}_{i}$. (Step 3) The next step was to standardize the resulting residual value $\left(Z_{i}\right)$ through the equation $Z_{i}=e_{i}/SD_{e\;\left(residual\right)}$, in which the residual value $\left(e_{i}\right)$ has to be divided by the residual standard deviation ($SD_{e}$). (Step 4) The final step consisted of seeking the exact score of the child, according to the standards by using the $\left(Z_{i}\right)$ value, the normal cumulative distribution function (in the case of normal distributions in the neuropsychological tasks) or the empirical cumulative distribution of residuals (in the case of nonnormal distributions in the neuropsychological tasks).

## 3. Results

Adjusted standards are offered for all scores, with the final multivariate models included in Appendix A. As can be seen in these tables, quadratic functions and interactions are common among all models.

Additionally, interactions between vulnerability and age or mean years of parental education were common. Figure 1 shows the shape of the predicted values in different neuropsychological tests, taking into account these interactions. For example, Figure 1a,b shows interactions between age and vulnerability. Specifically, in the observed ROCF copy scores (Figure 1a), in the case of younger children, the vulnerable group has lower scores than the nonvulnerable group. However, from approximately 10.5 years of age, the reverse is true, with the vulnerable children predicted to have higher scores. Predicted scores of M-WCST total errors are shown in Figure 1b. In this case, age shows a quadratic function and an interaction with vulnerability.

Hence, for vulnerable children, total errors are expected to progressively decrease and finally increase at approximately 13 years. In contrast, in the case of nonvulnerable children, the number of total errors is expected to progressively decrease with age. Considering, for example, the interactions between age and mean years of parental education (vulnerable: *M* = 6.47, *SD* = 4.48; nonvulnerable: *M* = 10.47, *SD* = 4.01) with the vulnerability factor, the Figures show the shape of these predicted values in different neuropsychological tests.

Figure 1c,d shows the interaction between parental education and vulnerability. Figure 1c shows the interference between mean years of parental education and vulnerability in the PPVT-III. Parents with higher levels of education may help to alleviate the effects of vulnerability in this receptive vocabulary test, with the children being able to reach the expected scores equal to those of nonvulnerable minors when the parents have approximately 21 years of education. As shown in Figure 1d, the mean years of parental education also interact with vulnerability in TAMV-I delayed response scores, with a mean of 8 years of education being a relevant point from which the tendency of the expected scores changed for the vulnerable group, and the nonvulnerable group was quite stable in this regard.

### Adjusted Standardization

In order to determine the percentiles, the personal data and Z_i_ were considered. The percentile score was given, taking into account only the significant variables in each task. Regression assumptions were tested in all cases. Multicollinearity was checked, using tolerance statistics (all values > 0.20) and Variance Inflation Factors (VIFs, all values < 10). Influential cases were detected using Cook’s distance, with the highest value being 0.120 in the case of M-WCST perseverative errors (in all cases < 1, [64]). The ROCF memory, verbal learning memory, SDMT, d2 concentration index, phonological verbal fluency (F, S, and A), semantic verbal fluency (fruits task) and Stroop test (color and word and interference) fulfilled the homoscedasticity assumption, which was assessed with the Levene test. Violations of the heteroscedasticity assumption were corrected in the other tests, using four standard deviations, one for each quartile, in the remainder of the measures. Normality was tested with the Kolmogorov–Smirnov test, with the scores in ROCF memory, TAMV-I immediate, SDMT, all measures of d2, PPVT-III, Token and Stroop color subtests being normally distributed. The lack of normality was corrected, using an empirical distribution.

The individual $Z_{i}$ score was included in the final model to obtain the individual percentile. The model for each task was included in an Excel file where the percentile appeared depending on participant traits. If percentile scores are needed from nonnormally distributed tasks, it is necessary to consult the empirical distribution tables (Appendix A). The standardized residual values are included in Appendix A.

## 4. Discussion

The objective of this study was to update ten neuropsychological tests for the pediatric Guatemalan population by adding more participants to the previous standards and including the condition of vulnerability in the update. The multivariate models showed that the interactions are significant in a relevant number of scores, especially between vulnerability and age or the mean parental education (MPE). Among others, this is relevant to the case of the PPVT-III, where the MPE seems to buffer the effect of vulnerability, with vulnerable children being able to reach their nonvulnerable counterparts’ scores when vulnerable parents have more years of education. This fact shows the influence that parental education has on children’s cognitive performance in some cognitive tasks.

The obtained models show that vulnerability was a significant factor in the vast majority of cognitive scores, especially in those related to language, attention and executive function tasks. These relations were noted in previous studies in which language and executive functions [65,66] were described as the cognitive domains most at risk in vulnerable children and adolescents. Other studies have also stated that attention [67] is a cognitive domain affected by low SES. In the present study, vulnerability was not significant in the multivariate models of ROCF memory, TAMV-I immediate memory, Stroop Color and Word, phonological fluency S and A subtest, or semantic fluency. That is, from the 26 scores, vulnerability was not a significant predictor in just eight of the scores. From the ten neuropsychological tests, the standards were updated by considering all measures. The model that could explain the most variance was the SDMT model (*R*^2^ = 0.678). The verbal memory recognition subtest model (*R*^2^ = 0.143) explained the lowest percentage of variance.

Adjusting neuropsychological standards is a normal practice in the field. Moreover, the inclusion of factors related to the living characteristics of the population that may affect cognitive performance is also a common practice in the neuropsychological field. The results offered in the present study fulfil a need identified in studies that have mentioned the importance of developing adjusted standards for minorities or specific groups of populations [7,10,12]. The methodology used in order to adjust the neuropsychological standards was used in other standardization studies [63,68] since it allows us to identify the predictable factors that may explain the task scores by controlling the multicollinearity, which is one of the assumptions that could be a cause for concern. Considering the normality or nonnormality of the data and including a method to determine the percentile score using the empirical distribution in each case makes the assessment process more accurate.

As was well described in the introduction, the importance of conducting the current study was realized after detecting recent studies that found relevant differences between neuropsychological performance in the case of vulnerable and nonvulnerable Guatemalan children [21]. The scoring process using standards was appropriately conducted in order to obtain the scores that truly correspond to children’s or adolescents’ cognitive performance. Since living in vulnerable conditions may have negative consequences for children and adolescents’ cognitive development, it is not appropriate to assess vulnerable populations with standards obtained by assessing nonvulnerable participants since this could lead to a type I error in which pathological scores could appear in neuropsychological tests when they actually do not exist [7].

The limitations of this study are mainly related to the concept of vulnerability. The authors of this study interpreted vulnerability conditions as experiencing socioeconomic hazards and being exposed to violence. There is no quantitative measure to define these variables when selecting the participants since the samples were selected based on the subjective criteria of contextual vulnerability. That is, schools from especially low socioeconomic and especially conflictive areas were selected. In order to choose these schools, although the authors were guided by professionals who work with these children and who are very familiar with their shortcomings and difficulties, it must be considered a limitation. Nonetheless, having a subjective criterion should be considered acceptable considering the relevance that this update has for children and adolescents from vulnerable contexts. Since there is evidence of the repercussions that poverty and violence have on children’s cognitive performance [23,69,70,71,72,73], this step should be considered a first move to go further in the field of child neuropsychology.

## 5. Conclusions

The neuropsychology field needs to be supported by adequate methods of interpreting neuropsychological scores. The main idea we sought to convey is that the adaptation of neuropsychological tests should go beyond mere translation from one language to another or the changing of items to be understood by a certain group of the population. It is important to understand how personal context affects individual performance, especially in the case of psychological and neuropsychological tests.

Clinical practice needs to have adequate measures to avoid the use of inadequate interpretation of data on neuropsychological measures. Thus, Latin American professionals specializing in neuropsychology have severe limitations when finding proper standards to apply to their populations [4]. In the case of Guatemala, a country with high poverty and great cultural diversity, this work allows us to offer data that can help in the practice of child neuropsychology. Additionally, due to the child poverty rates in some Latin American countries, the vulnerability variable is a key factor that should be considered in clinical practice. In this manuscript, the authors have offered the next step that needs to be taken for solving these assessment deficiencies.

The measurement of cognitive performance in children and adolescents needs adaptation to the different issues that these disadvantaged populations may experience since their contextual hazards interfere with the way in which they receive and manage information. In the case of the assessment of minorities, it would be interesting to extend the use of culture-free neuropsychological tests in order to faithfully measure the variables of interest. Moreover, it would be interesting to go further than the standards themselves and to analyze the most used neuropsychological tests in order to see their suitability for different groups.

This study contributes to neuropsychology by offering solutions to issues that are considerably a cause for concern for the United Nations. Including vulnerability as a variable to consider in the standardization of neuropsychological tests makes this reality visible. By possessing suitable standards, we will be able to conduct fairer assessments with which to help in the early detection of special needs and developmental deficits in this disadvantaged population. In future studies, the authors of this article plan to extend this methodology in order to update other vulnerable populations’ standards. These vulnerability-fitted standards can promote more accurate and relevant interventions, and contribute to the creation and implementation of policies that, based on the Sustainable Developmental Goals (SDGs) of the 2030 Agenda, may help to offer better developmental opportunities to thousands of children and adolescents around the world.

## Figures and Tables

**Figure 1 brainsci-11-00842-f001:**
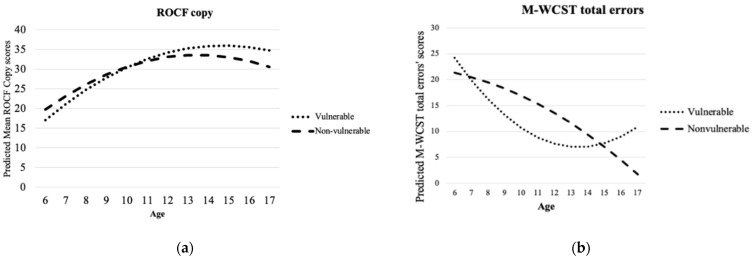
Interactions included in models. (**a**) Predicted ROCF copy subtest scores; (**b**) predicted M-WCST total errors scores; (**c**) predicted PPVT-III scores; (**d**) predicted TAMV-I delayed responses’ scores.

## Data Availability

The data presented in this study are available on request from the corresponding author. The data are not publicly available due to the sensitivity of the data collected from underage participants.

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
