# Peer review of "Normative Data for Ten Neuropsychological Tests for the Guatemalan Pediatric Population Updated to Account for Vulnerability"

_brainsci, 2021, doi:10.3390/brainsci11070842_

Round 1
Reviewer 1 Report
The aim of this article was to update the standardized scores for ten neuropsychological tests commonly used in Guatemala considering vulnerability. As written in the article the standard scores were created using multiple linear regressions and standard deviations from residual values. The predictors included were the following: age, age2, mean level of parental education, MLPE2, gender, and vulnerability, as well as their interaction.
My comments to the article are as follows:
- The manuscript has been submitted to BrainSciences and has a Children form in the header. Please verify.
- I suggest updating the title of the work. This data update should be at the end of this title rather than the beginning. In my opinion, it doesn't fit right now.
- As part of the Introduction, I propose to provide a broader background, also referring to the fact of research on stress, which is significant in the neuropsychological field. For example, I suggest referring to: The Impact of Different Sounds on Stress Level in the Context of EEG, Cardiac Measures and Subjective Stress Level: A Pilot Study, Brain Sciences from 2020.
- I am asking you for detailed information on the selection of such and not another study group, in particular in terms of its size.
- I propose to describe the mathematical formula in line 373 with a number and refer to it in the text of the article.
- The data presented in the Results section is modest. I propose to expand this section of the article. The graphs in Fig. 1 are hardly visible. Perhaps it is worth dividing them. I also propose to complete the units of measurement in this regard.
- As part of the Conclusions, I propose to describe the plans for the future in the scope of the conducted research.
Author Response
Changes applied related to comments by reviewer 1 will appear into the manuscript in yellow
Reviewer 1
The aim of this article was to update the standardized scores for ten neuropsychological tests commonly used in Guatemala considering vulnerability. As written in the article the standard scores were created using multiple linear regressions and standard deviations from residual values. The predictors included were the following: age, age2, mean level of parental education, MLPE2, gender, and vulnerability, as well as their interaction.
My comments to the article are as follows:
- The manuscript has been submitted to BrainSciences and has a Children form in the header. Please verify.
We apologize for this oversite. This use has now been corrected.
- I suggest updating the title of the work. This data update should be at the end of this title rather than the beginning. In my opinion, it doesn't fit right now.
We thank the reviewer for this suggestion. The title has been modified.
- As part of the Introduction, I propose to provide a broader background, also referring to the fact of research on stress, which is significant in the neuropsychological field. For example, I suggest referring to: The Impact of Different Sounds on Stress Level in the Context of EEG, Cardiac Measures and Subjective Stress Level: A Pilot Study, Brain Sciences from 2020.
We thank the reviewer for this suggestion. The introduction has been revised to provide a broader background and the article indicated has been included in the new version of the manuscript. The majority of these updates can be found above line 111
- I am asking you for detailed information on the selection of such and not another study group, in particular in terms of its size.
The sample was selected by convenience, since this study was conducted in the frame of the project descripted in the Funding section. This information has been included in the manuscript around the line 165.
- I propose to describe the mathematical formula in line 373 with a number and refer to it in the text of the article.
A description of the formula has been included.
- The data presented in the Results section is modest. I propose to expand this section of the article. The graphs in Fig. 1 are hardly visible. Perhaps it is worth dividing them. I also propose to complete the units of measurement in this regard.
We appreciate your comment. Regarding the results section, it is important to consider that it includes the information provided just after the section’s title and the subsection 3.1 “Adjusted standardization”. In the first part we have explained some examples of the most relevant findings. In the second part, we have offered an explanation about the findings regarding linear regression assumptions. From our point of view all this information is complete and pertinent to understand the relevant findings in this study.
The graphs have been modified in order to improve their visibility and understanding. The units of measurement are quite different because they represent scores in different neuropsychological tasks. The graphs have been grouped depending on linear regressions results. Graphs 1.a and 1.b correspond to scores whose interactions with vulnerability have been significant with sample ages (from 6 to 17). On the other hand, graphs 1.c and 1.d show scores whose interactions with vulnerability have been significant with sample’s mean year of parental education. This section has been rewritten, and we believe it now offers a clearer and more detailed example of some of the most important results.
- As part of the Conclusions, I propose to describe the plans for the future in the scope of the conducted research.
Our future plans have been included into the conclusions section.
Reviewer 2 Report
Thank you for the opportunity to review this work. Generally speaking, it’s concerned with an important topic. In its current state, two major concerns arise: the construct of vulnerability needs to be clearly defined (the authors included a short discussion about that as a limitation). Only with a clear idea of what this means, how to measure and quantify it, the value of this work evolves. Currently, the idea behind the construct is briefly mentioned, but a clear definition is missing, and hence the grouping of participants into the non-/vulnerable groups is comprehensible. Secondly, the authors seek to update the standards of established questionnaires and tests. Similarly, the term “update” in this context is undefined, and it remains unclear how the tests mentioned are now updated in the discussed regard. While the idea is great and the study carried out seems reasonable, both the applicability of the findings and the statistical reporting remain vague and should be clarified.Following this, I’ll discuss a couple of passages in detail: “It is easy to imagine that as vulnerable children have special conditions that they face on a daily basis (e.g., undernutrition, crowded households, lack of stimulation, difficulties in family attachment, possible trauma, etc.), they should not be measured with the same scales as children without these living conditions.” (l128-l131, introduction) - This generally depends on the impact of vulnerability on the outcome measures of these tests. - Furthermore, a discussion on the outcome constructs is needed: if a vulnerability score infers the construct “performance” (in a given context), the construct (performance) is not as clearly defined as assumed. Hence, an alternative to introducing a mediating variable (vulnerability) would be establishing an environment-free construct and discussing the test at hand (I acknowledge that this might be out of focus for this work. However, I think the authors could discuss this). “Thus, measuring children’s cognitive capacities with scales created using nonvulnerable populations leaves thousands of children and adolescents in a disadvantaged situation as they are being compared with a group that does not belong to their normal context.” (l131-l134, introduction) - I think this is a great and vital point. The methods section starts with a statement regarding the participants, 203 nonvulnerable and 144 vulnerable participants (l153 & l154). However, the authors did not define the vulnerability construct before. L125-l128 briefly discusses influencing factors to this construct, but it remains unclear how these aspects were operationalized and put together to form a binary score (non-/vulnerable). Furthermore, a discussion on the binary nature of this variable would be desirable: why is this not expressed as a continuous variable? At this point, it becomes clear that I don’t have particular expertise regarding this dimension. However, to make the manuscript accessible to more researchers from different disciplines and backgrounds, it’s important to elaborate on the construct. It’s unclear what “updating the standards” means (l147, l155). I assume the authors suggest computing new average scores for the population discussed, including non-/vulnerable participants. In l155 the authors define the non-/vulnerable groups. It seems likely that these groups differ in their exposure to risks (among others) and show significant differences regarding income, education, and social status, to name a few. To understand if these are potentially confounding factors or are exactly what the vulnerability construct should capture, a thorough definition of the construct is needed (and currently missing). E.g., parental education seems not to be included in the vulnerability construct (l405) but will most likely show significant differences between an urban and a rural area. The sentence regarding income starting in l161 seems detached from the rest of the paragraph. With a mean number of years of parental education of 8.8 years and an SD of 4.7 years, the. The x-axis of figure 1.c seems problematic. How are such large numbers of years of education possible (e.g., 20 years)? And how representative are they? How many participants did you sample with such a high number that it’s justifiable to extend the figure to these values? (l408). I suggest adding ribbons to the graphs to indicate the width of acceptance, e.g., SD or 95% CIs. The results should include general goodness-of-fit measures for the regression models. A statistician should review the results. Additional, minor issues: - L92: “these authors” potentially unclear who “these” refers to (Arango-Lasprilla et al. [4]) - L98: missing whitespace “published[14].” - L189: missing whitespace “population[28].” - L237: highlighted “>= 12 years old” but l243 did not have such a highlighting - L259: missing whitespace “attention[40]” - L358: p-values below .001 should (probably) be reported as such (instead of < .0001) - L468: potentially a word missing after “is also a common practice in neuropsychological.”
Author Response
Changes applied related to comments by reviewer 2 will appear into the manuscript in green
Reviewer 2
1.Thank you for the opportunity to review this work. Generally speaking, it’s concerned with an important topic. In its current state, two major concerns arise: the construct of vulnerability needs to be clearly defined (the authors included a short discussion about that as a limitation). Only with a clear idea of what this means, how to measure and quantify it, the value of this work evolves. Currently, the idea behind the construct is briefly mentioned, but a clear definition is missing, and hence the grouping of participants into the non-/vulnerable groups is comprehensible. Secondly, the authors seek to update the standards of established questionnaires and tests. Similarly, the term “update” in this context is undefined, and it remains unclear how the tests mentioned are now updated in the discussed regard.
The project from which this study is based, is indeed based on a first project that aimed to create standards for neuropsychological tasks in Latin-American and Spanish infant population. So, first a research group conducted a project to create standards for 10 neuropsychological tests for Latin-American and Spanish populations. In the samples used to create the norms there was a higher number of younger children (around 6 years old) than adolescents. Subsequently, we conducted a project focused on Guatemalan population which is characterized by high violence and poverty rates. The term “update” in this case stands for the fact of upgrading the previously existing data for Guatemalan population by adding more adolescents to the sample and by considering the vulnerability of the sample. In the text, this is explained around line 94 (Thus, in 2017…). We have clarified the term “update” around the line 149 and this information is completed by the study’s objective (line 163).
2.While the idea is great and the study carried out seems reasonable, both the applicability of the findings and the statistical reporting remain vague and should be clarified.
Following this, I’ll discuss a couple of passages in detail:
“It is easy to imagine that as vulnerable children have special conditions that they face on a daily basis (e.g., undernutrition, crowded households, lack of stimulation, difficulties in family attachment, possible trauma, etc.), they should not be measured with the same scales as children without these living conditions.” (l128-l131, introduction)
- This generally depends on the impact of vulnerability on the outcome measures of these tests.
As we pointed out in the introduction (lines 113 to 132), the consequences of growing up in such conditions have shown to negatively impact on brain’s structure and function.
- - Furthermore, a discussion on the outcome constructs is needed: if a vulnerability score infers the construct “performance” (in a given context), the construct (performance) is not as clearly defined as assumed.
We appreciate your comment. As part of the introduction, we have explained that factors related to vulnerability, which negatively impact to cognition. In methods section the reader can see the compendium of tests that have been used in order to assess different cognitive processes. In the description of each test the cognitive processes assessed are included. We have referenced a study in which the outcomes are defined and discussed considering their relation to vulnerability (around line 131).
- Hence, an alternative to introducing a mediating variable (vulnerability) would be establishing an environment-free construct and discussing the test at hand (I acknowledge that this might be out of focus for this work. However, I think the authors could discuss this).
We totally agree with you. It would be desirable to conduct a study in which the pertinence of each neuropsychological tests would be analyzed in terms of the population to be assessed. Since this is a really interesting and relevant point to discuss, as you suggested, we have added this consideration into the conclusions’ section.
- “Thus, measuring children’s cognitive capacities with scales created using nonvulnerable populations leaves thousands of children and adolescents in a disadvantaged situation as they are being compared with a group that does not belong to their normal context.” (l131-l134, introduction) - I think this is a great and vital point.
Thanks you. This is the main idea on which this manuscript is based.
- The methods section starts with a statement regarding the participants, 203 nonvulnerable and 144 vulnerable participants (l153 & l154). However, the authors did not define the vulnerability construct before.
Because we do not use a quantitative variable to define the sample, we have used a qualitative definition provided by the Royal Spanish Academy, and we have related this definition to the assessed sample (This information is highlighted in the line 135). We have extended this explanation (line 141-146).
- L125-l128 briefly discusses influencing factors to this construct, but it remains unclear how these aspects were operationalized and put together to form a binary score (non-/vulnerable).
This point is now explained. The sample was selected by convenience (line 182). We based the selection on experts’ advices, as we have pointed out into the text.
- Furthermore, a discussion on the binary nature of this variable would be desirable: why is this not expressed as a continuous variable? At this point, it becomes clear that I don’t have particular expertise regarding this dimension. However, to make the manuscript accessible to more researchers from different disciplines and backgrounds, it’s important to elaborate on the construct.
It is difficult to determine a cut-off point since which we can determine a child is vulnerable or not in the contexts this sample was taken. We are talking about especially at-risk contexts, where the scarcity is visible and the criminal gangs are prevalent in the surrounding neighborhoods. We have expanded this explanation and the justification to choose a qualitative concept to approach this construct (141 and 182).
- It’s unclear what “updating the standards” means (l147, l155). I assume the authors suggest computing new average scores for the population discussed, including non-/vulnerable participants.
We understand that this concept could be confusing. We have addressed this point in the “aim of the study paragraph above the line 167.
- In l155 the authors define the non-/vulnerable groups. It seems likely that these groups differ in their exposure to risks (among others) and show significant differences regarding income, education, and social status, to name a few. To understand if these are potentially confounding factors or are exactly what the vulnerability construct should capture, a thorough definition of the construct is needed (and currently missing). E.g., parental education seems not to be included in 11. The vulnerability construct (l405) but will most likely show significant differences between an urban and a rural area.
We do agree with you. We have extended the explanation why we have chosen this concept in the line 136.
- The sentence regarding income starting in l161 seems detached from the rest of the paragraph.
We have modified the sentence in order to make it clearer.
- With a mean number of years of parental education of 8.8 years and an SD of 4.7 years, the. The x-axis of figure 1.c seems problematic. How are such large numbers of years of education possible (e.g., 20 years)? And how representative are they?
The x-axis is built depending on the obtained scores in the Formula 1. We have included those values in order to show what could happen in case there is a family which mean years of parental education is 17 for instance, since there are cases of this value. The values reached 20 because there were families which the mean years of parental education was 20. Now, information about mean years of parental SD is included in the results section (around line 433)
- How many participants did you sample with such a high number that it’s justifiable to extend the figure to these values? (l408). I suggest adding ribbons to the graphs to indicate the width of acceptance, e.g., SD or 95% CIs.
We appreciate your suggestion. We have not included the confident intervals since each graph represents just an element of the regression models. This is, graphs do not include the complete regression models. Nonetheless, in case you think it is convenient to include this information we can provide this information.
We have included though, the SD information relevant to interpret the results into the results section (line 433 aprox.)
- The results should include general goodness-of-fit measures for the regression models. A statistician should review the results.
We appreciate your comment. One of the authors of this manuscript is expert on the methodology applied. We have offered the results pertinent data to test the linear regression’s assumptions (around line 440).
- Additional, minor issues: - L92: “these authors” potentially unclear who “these” refers to (Arango-Lasprilla et al. [4]) - L98: missing whitespace “published [14].” - L189: missing whitespace “population [28].” - L237: highlighted “>= 12 years old” but l243 did not have such a highlighting - L259: missing whitespace “attention [40]” - L358: p-values below .001 should (probably) be reported as such (instead of < .0001) - L468: potentially a word missing after “is also a common practice in neuropsychological.
We have applied those changes, thank you.
Reviewer 3 Report
Suggestions:
- Clearer communication in the abstract. It is hard to understand what age2 mean, without reading full-text.
- Introduction, lines 129-134. Is the problem that similar test cannot be used, or that specific data is needed? In this publication authors provides regression scores for adjustment of normative data, and do not suggest any test modifications, thus, I would suggest changing argumentation accordingly.
- Methods. Age range of the children sample is wide for a children population. Different normative data should be used for children at age of 6 and 17. Please provide more detailed description of the sample, including distribution of children between smaller age groups.
- Information regarding informed consent is not clear. Who gave informed consent for children participation?
- Description of parents sample is not provided.
- Table S11 is not clear. Please specify values in the column "predictive values". It is hard to understand how are they related to the column "Neuropsychological test".
Author Response
Changes applied related to comments by reviewer 3 will appear into the manuscript in blue
Reviewer 3
- Clearer communication in the abstract. It is hard to understand what age2 mean, without reading full-text.
Thank you for your comment. We have applied this modification.
- Introduction, lines 129-134. Is the problem that similar test cannot be used, or that specific data is needed? In this publication authors provides regression scores for adjustment of normative data, and do not suggest any test modifications, thus, I would suggest changing argumentation accordingly.
We have corrected the sentence so now the information is clearer.
- Methods. Age range of the children sample is wide for a children population. Different normative data should be used for children at age of 6 and 17. Please provide more detailed description of the sample, including distribution of children between smaller age groups.
Normative data depend on children’s ages. Table S1 with two age ranges information (6-11 and 12-17 years) has been included in the Supplementary material 1.
- Information regarding informed consent is not clear. Who gave informed consent for children participation?
This piece of information is now clear around line 193
- Description of parents’ sample is not provided.
Relevant information about parent characteristics regarding this study is provided in Materials and methods section, around line 171.
- Table S11 is not clear. Please specify values in the column "predictive values". It is hard to understand how are they related to the column "Neuropsychological test".
We appreciate your comment. A footnote has been included in order to clarify this information
Round 2
Reviewer 1 Report
Dear Authors,
Thank you for making changes to the article. Thank you also for the responses to my questions / concerns.
I recommend the article for publication.
Reviewer 2 Report
The authors improved the manuscript significantly and incorporated all suggestions made. Thank you!
Minor issue: Typo in l101 “The normative data produced by f this process…”